# Characteristics, Health Risk Assessment, and Transfer Model of Heavy Metals in the Soil—Food Chain in Cultivated Land in Karst

**DOI:** 10.3390/foods11182802

**Published:** 2022-09-11

**Authors:** Liyu Yang, Pan Wu, Wentao Yang

**Affiliations:** 1College of Resource and Environmental Engineering, Guizhou University, Guiyang 550025, China; 2Key Laboratory of Karst Geological Resources and Environment, Ministry of Education, Guizhou University, Guiyang 550025, China; 3Guizhou Karst Environmental Ecosystems Observation and Research Station, Ministry of Education, Guizhou University, Guiyang 550025, China

**Keywords:** soil, heavy metals, foods, health risk assessment, transfer model

## Abstract

Heavy metal(loid) contamination of farmland is a crucial agri−environmental problem that threatens food safety and human health. In this study, we examined the contamination levels of heavy metals (As, Pb, Cd, Hg, Cr) in farmland and foods (rice, maize, and cabbage) in the core of Asia’s largest karst region and assessed the potential health risks of consumption of these three foods. In addition, we developed a predictive transfer model of heavy metals in the soil−food chain through multiple regression equations. The results reveal that the soil heavy metals in the study area showed high accumulation characteristics, and the average concentration exceeded the national background value by 1.6−130 times, among which Cd pollution was the most serious. The order of contamination of the three soils in the study area was cabbage land > maize land > rice land. The order of potential risk of toxic elements in all three soils was Cd > Hg > As > Pb > Cr. The results of the risk assessment of agricultural consumption indicated a high carcinogenic and noncarcinogenic risk for the local population. The top contributor to carcinogenic risk was Cr, followed by As. Cd is the major noncarcinogenic contributor in maize and cabbage, and the noncarcinogenic contribution in rice is mainly caused by As. The risk was higher in children than in adults and was the highest for rice consumption. In addition, the predictive transfer model showed that the Cd levels in the three foods showed sufficient predictability and reasonable simulations of Cd concentrations in rice, maize, and cabbage throughout the study area. It could allow decision-making on the need for remediation strategies to reduce the risk of metal contamination of agricultural land in potentially high−risk areas of karst.

## 1. Introduction

Soil is the essential ecosystem for human survival and development, as well as an important resource for people to carry out agricultural production activities [1,2,3]. With the rapid expansion of industrialization, heavy metal contamination of soil has now become a severe environmental problem [4,5]. Heavy metals are not easily decomposed by microorganisms in the soil and are persistent and toxic [6,7]. Heavy metals tend to accumulate in the soil and are enriched by plant uptake into organisms, which ultimately endangers human health [8,9]. For example, Cd and Cr can cause damage to human kidneys and liver [10,11]; even if the content of Hg in the water environment is extremely low, it can be harmful to the human central nervous, respiratory, and cardiovascular systems [12]; As is teratogenic and can cause cancer and increase the incidence of coronary heart disease [13]; Pb can affect red blood cells and brain, kidney, and nervous system functions in the human body [14].

Guizhou Province of China has the largest karst core region in Asia, with a fragile ecological environment [15]. Soils in karst areas are poor, thinly weathered, highly porous, and unevenly distributed, which leads to weak carrying capacity and flexible migration of pollutants [16]. Guizhou Province belongs to a high geological background area for heavy metals, where the soil Cd background value is 0.66 mg kg^−^^1^, which is significantly higher than the national Cd background value of 0.097 mg kg^−^^1^ [17]. Meanwhile, it is abundant in mineral resources, with more than 110 types of minerals, including mercury, barite, coal, antimony, and gold, identified [18]. The heavy metals produced by a large number of mining activities have polluted the surrounding farmland and atmosphere, causing serious ecological and environmental problems [19]. Mining, metallurgy, and other human activities are superimposed on high geochemical background values of heavy metals, which exacerbates the accumulation of heavy metals in karst agricultural land soils and makes the safety of regional agricultural products more prominent [20]. In recent years, health risk evaluation studies of heavy metal pollution in soils of karst areas have received much attention [21,22,23]. Wang et al. [24] studied a typical small watershed contaminated by a zinc powder mill in northwestern Guizhou Province, where there was a serious contamination of Cd, As, Pb, Cu, and Zn in the soil and the ecological risk was considered high. Xiao et al. [25] studied the spatial distribution, ecological risk, and possible sources of nine heavy metals in the soil of Huishui County, a karst area. The study found that the potential ecological risk of Hg and Cd was the highest, and the main sources were attributed to anthropogenic causes. Huang et al. [26] used principal component analysis to analyze the sources of heavy metals in the largest karst wetland in South China (Hui County) and found that the main source could be the weathering of parent carbonate rocks. A total of 113 soil samples were collected in the karst−bearing area of Hengxian, Guangxi, China. Jia et al. [27] conducted a redundancy analysis of 18 influencing factors in the soil and found that natural sources were the main source of heavy metals in the study area.

However, most of the previous studies have emphasized heavy metal contamination in soils of karst areas and its sources [19,28] and often neglected heavy metals in foods, and their relationships have not been fully explored [29,30,31,32]. Rice (*Oryza sativa* L.), maize (*Zea mays* L.), and cabbage (*Brassica oleracea* L. *var. capitata* L.) are the main daily foods in the study area, and previous surveys have shown that there may be potential ecological risks [33,34,35]. Meanwhile, this study attempted to develop a new soil−crop heavy metal transport model by establishing multiple regression equations for total metal concentrations in soil, soil pH, and metal concentrations in crops. This study provided a valuable tool for the evaluation of heavy metal health risk in farmland soils and prediction of heavy metal transformation in soil−crop chains; it can make decisions on the necessity of remediation strategies to reduce the risk of metal contamination in farmland in potentially high−risk areas of karst.

The main objectives of this study, using the mountainous farmland in Shuicheng County as the study area, were to (1) determine the contents and contamination levels of heavy metals (Cd, Hg, As, Pb, and Cr) in agricultural soils in the study area; (2) analyze the content characteristics of heavy metals in rice, maize, and cabbage and assess the potential health risks to adults and children from consuming the three crops; and (3) develop a set of prediction models for the accumulation capacity of heavy metals in soil and crops through multiple regression equations.

## 2. Materials and Methods

### 2.1. Investigation Areas

The study was conducted in Shuicheng District of Guizhou Province (104°33′–105°15′ N, 26°03′–26°55′ E) from June 2019 to October 2020. The investigation area belongs to the subtropical humid monsoon climate zone, with average annual temperature of 12.4 °C and a significant rainy season (annual average precipitation 1100 mm). Moreover, the area has an annual sunshine of 1300–1500 h and an annual frost−free period of 250 days suitable for producing food and cash crops [33].

### 2.2. Sampling and Sample Analysis

Through ArcGIS software (Version 10.8, Environmental Systems Research Institute, Inc., Redlands, CA, USA), 160 groups of soil—crop synoptic monitoring sites were randomly deployed in the main crop−growing areas, namely 22 groups of rice, 103 groups of maize, and 35 groups of cabbage. The edible parts of the crops were collected and placed in polyethylene mesh bags, whereas the topsoil layer (0–20 cm) corresponding to the crop rhizome layer was collected with a wooden shovel. Each pair of samples was a mixture of three parallel samples from the longest diagonal of the actual field in the area where the setup point was located. After the soil and crop samples were brought back to the laboratory, the soil samples were naturally dried, removed from the rootstock debris and ground, passed through 0.149 mm nylon mesh sieves, and stored in separate packs at room temperature. The edible part of the crop samples was washed with tap water, rinsed with deionized water 3–5 times, blanched at 105 °C for 30 min, dried at 75 °C until constant weight, broken, ground, passed through a 0.149 mm nylon sieve, and then stored in separate packs at room temperature. The pH value of the soil was tested with a pH meter (pH–3c, INESA Scientific, Shanghai, China), and the water–soil weight ratio was 2.5:1 (*w*/*w*). A HNO_3_–HClO_4_ solution was used to digest the agricultural crops. The HNO_3_–HCl–HF–HClO_4_ solution was used to digest the soil. ω(Cr), ω(Cd), and ω(Pb) in the digest were determined by inductively coupled plasma–mass spectrometry (ICP–MS, Thermo Fisher Scientific, Waltham, MA, USA), and ω(As) and ω(Hg) in the digest were determined by hydride–atomic fluorescence spectrometry (HG–AFS) [36].

### 2.3. Contamination Evaluation and Health Risk Assessment Model

The degree of heavy metal contamination in soil is analyzed using the geo−accumulation index (I_geo_), which is calculated as follows:(1)$$I_{\mathrm{geo}}={\mathrm{Log}}_{2}\frac{C_{s}}{1.5\times B_{n}}$$
where *C_s_* is the heavy metal test concentration, and *B_n_* is the background value of heavy metals in Guizhou Province (Cd 0.66 mg kg^−1^, Hg 0.11 mg kg^−1^, As 20.00 mg kg^−1^, Pb 35.20 mg kg^−1^, and Cr 95.90 mg kg^−1^) [37]. The grading of the geo−accumulation index and the pollution status were divided into 7 levels, as detailed in Table 1 [38].

The potential ecological risk index [10] proposed by the Swedish scholar Hakanson is based on the physical and chemical properties of heavy metals and the interaction of the environment and uses a comparable equivalence property index grading method to evaluate its calculation formula as follows:(2)$$E_{r}=\mathrm{TF}\times \mathrm{CF}=\mathrm{TF}\times \frac{C_{i}}{C_{B}}$$
(3)$$R=\sum E_{r}$$
where E_*r*_ is the potential ecological risk index; TF is the toxicity effect factor, and the five heavy metals’ TF are 30 for Cd, 40 for Hg, 10 for As, 5 for Pb, and 2 for Cr; CF is the individual heavy metal pollution index; *C_i_* is the measured concentration of heavy metal “*i*” in soil; *C_B_* is the sediment background concentration (reference concentration); and R is the combined potential ecological risk index. The gradings of E_*r*_ and R are shown in Table 2.

The potential of adverse health effects from human consumption of heavy metals was evaluated by consuming rice, maize, and cabbage. This research used the general exposure equation for health risk assessment provided by the US Environmental Protection Agency. The metal estimated daily intake (EDI) (mg/(kg d^−1^)) formula is used to calculate human exposure levels. The EDI should be calculated using Equation (4) [39]:(4)$${\mathrm{EDI}}_{i}=\frac{C_{i}\text{×}\mathrm{IR}\text{×}\mathrm{EF}\text{×}\mathrm{ED}}{\mathrm{BW}\text{×}\mathrm{AT}}$$
where *C_i_* is the content of heavy metals “*i*” in foods (mg kg^−1^), and IR is the ingestion rate (kg d^−1^) (0.14 kg d^−1^ and 0.32 kg d^−1^ for rice: children and adults; 0.100 kg d^−1^ and 0.150 kg d^−1^ for maize: children and adults; and 0.100 kg d^−1^ and 0.150 kg d^−1^ for cabbage: children and adults, respectively) according to the results of our onsite survey. EF is the exposure frequency (d/year), taken as 365 days/year, ED is the exposure duration (children: 10 years; adults: 30 years), and BW is the weight (kg) of the exposed individual (children: 16 kg; adults: 61.75 kg). AT is the average exposure time to heavy metals and is equal to ED × 365 d/year.

The hazard quotient (HQ) is used to evaluate the noncarcinogenic risk of a single heavy metal after food intake [40]:(5)$$\mathrm{THQ}=\sum {\mathrm{HQ}}_{i}=\sum \frac{{\mathrm{EDI}}_{i}}{{\mathrm{RfD}}_{i}}$$
where *RfD* is the reference dose (mg/(kg d^−1^)), and the values are Cr: 3 × 10^−3^, As: 0.3 × 10^−3^, Cd: 1 × 10^−3^, and Pb: 3.5 × 10^−3^. HQ > 1 indicates that the pollutant causes health risk to humans, and HQ < 1 indicates that the pollutant causes negligible health risk to humans.

Carcinogenic risk due to ingestion of heavy metals from foods can be assessed based on the carcinogenic risk (CR), which is calculated as follows [41]:(6)$$\mathrm{TCR}=\sum {\mathrm{CR}}_{i}=\sum {\mathrm{EDI}}_{i}\times {\mathrm{SR}}_{i}$$
where SF is the cancer slope factor, and the values are Cr: 0.50; As: 1.50; Cd: 0.0038; and Pb: 00085. A carcinogenic risk index (CR) below 1 × 10^−6^ indicates negligible carcinogenic risk, above 1 × 10^−4^ indicates unacceptable carcinogenic risk, and in between indicates the existence of an acceptable carcinogenic risk.

### 2.4. Prediction Model for Metal Content in Foods

To predict the heavy metal concentrations in food, we modeled the soil−producing data in simulations. Some data sets showed very high concentrations of heavy metals in soil and low concentrations of metals in edible parts of foods. These data sets were outliers and were not used in the model simulations. After removing the outlier data sets, the data were used in the model simulations. The concentration of heavy metals in food is mainly determined by the soil pH and the concentration of heavy metals in the soil [42]. Based on previous studies, the following equations were used in the prediction model:(7)$${\mathrm{Log}C}_{f}={a+b\text{×}\mathrm{pH}+c\text{×}C}_{S}$$
where *C_f_* and *C_s_* represent the heavy metal concentrations in food and soil, respectively (mg kg^−1^); a is a constant value in mg kg^−1^, and pH is the soil pH value; b and c are the slopes of the soil pH and soil heavy metal concentration, respectively. The predicted results depend on the *p* and R^2^ values.

### 2.5. Quality Control and Statistics

All chemicals were of reagent grade, and deionized water was used in all experiments. All glassware and utensils were cleaned, soaked in nitric acid solution (10% *v*/*v*) overnight, rinsed with deionized water, and dried before use. The National Standard Materials of soil (GBW07404a (GSS−4a)) and Maize (GBW10012 (GSB−3)) were used during the analysis for quality control. The recovery rates of total heavy metals were 95.7−108.9% and 93.1−107.3%, respectively.

All data were analyzed by Excel 2021. The data are expressed as the mean ± standard deviation (*n* = 3). All figures were processed using OriginPro software (Version 2021, OriginLab Corp., Northampton, NC, USA).

## 3. Results

### 3.1. Soil pH Values and Heavy Metal Concentrations

The pH values and heavy metal concentrations of the soil in the study area are presented in Table 3. The soil in the study region was neutral and slightly acidic, and the median values of soil pH for rice, maize, and cabbage were 6.05, 6.3, and 6.12, respectively. The mean values were 6.03, 6.35, and 6.12, respectively, with ranges of 4.67–7.47, 4.53–8.09, and 4.35–7.77, respectively. The coefficients of variation of pH for rice, maize, and cabbage soils were 0.12, 0.15, and 0.15, respectively, which could be interpreted as low variability (CV < 0.5). In addition, the skewness coefficients of the pH values of all three soil types were small (>−0.5, <0.5), and the kurtosis coefficients were in the range of −2 to 2, implying a normal distribution. These results indicate negligible anthropogenic influences on the pH of the three soil types.

The concentrations of heavy metals (Cd, Hg, As, Pb, Cr) in the three types of soils showed high accumulation characteristics, among which Cd pollution was the most serious. The ranges of ω(Cd), ω(Hg), ω(As), ω(Pb), and ω(Cr) in maize soils were 0.3–15.5, 0.06–2.87, 2.74–532, 18.50–696, and 61.3–404 mg kg^−1^, and their mean values were 2.96, 0.25, 31.75, 71.69, and 168.98 mg kg^−1^, respectively. The order of the coefficients of variation was As > Pb > Hg > Cd > Cr, where the variation types of As, Pb, and Hg were strong variants (CV > 1.0). The concentrations of ω(Cd), ω(Hg), ω(As), ω(Pb), and ω(Cr) in rice soils were 0.26–3.55, 0.04–0.32, 3.23–51.9, 18.2–102, and 52.3–216 mg kg^−1^, and their mean values were 1.17, 0.11, 13.57, 39.01, and 137.57 mg kg^−1^, respectively. The order of the coefficient of variation was As > Cd > Hg > Pb > Cr, where As had the highest coefficient of variation and could be interpreted as strong (CV > 1). The concentrations of ω(Cd), ω(Hg), ω(As), ω(Pb), and ω(Cr) in cabbage soils were 4.75–7.77, 1.06–64.7, 0.05–0.32, 3.78–81.7, and 30–1864 mg kg^−1^, respectively. The mean values were 6.13, 13.01, 0.14, 24.77, and 320.6 mg kg^−1^, respectively. The order of the coefficients of variation was Pb > Cd > As > Hg > Cr, where the variation types of Pb and Cd were strong variants (CV > 1.0).

Compared with the national soil background values (NBGV), the average levels of Cd, Hg, As, Pb, and Cr in the three soil types in the study area exceeded the NBGV by 1.6–11.7, 2.8–29.6, and 2.1–130 times, respectively. It is worth mentioning that the percentage of Cd points exceeding BGV in the three soils was 100%, which was higher than 98.13% for Cr, 91.25% for Hg, 89.38% for Pb, and 69.38% for As. Except for Cd in rice soils (CV = 0.49), Cd, As, and Pb in all three soils reached moderate or strong variability levels (CV > 0.5), implying a high spatial variability and a strong impact of local anthropogenic activities, relatively [43,44]. The skewness coefficients of Cd, Hg, As, Pb, and Cr for maize, rice, and cabbage soils were 0.93–8.08, 0.14–2.16, and 0.25–2.63, respectively, indicating that those toxins were minor positively skewed, with a larger number of values clustered in the low range. The kurtosis coefficients of Cd, Hg, As, and Pb in maize soils were large (>8, not consistent with a normal distribution), indicating that the four heavy metals clustered around their mean values. The relatively small kurtosis of each heavy metal in the other soils indicates that the data are scattered.

### 3.2. Heavy Metals in Foods

The statistical results of the heavy metal concentrations of the three foods in the study area are shown in Table 4. Among them, 22 rice samples had Hg below the limit of detection of the instrument (LOD_Hg_ = 0.16 μg L^−1^), and Pb was also not detected in seven rice samples (LOD_Pb_ = 0.02 μg L^−1^). The mean values of ω(Cd), ω(As), ω(Pb), and ω(Cr) in the remaining rice samples were 0.071, 0.052, 0.020, and 0.355 mg kg^−1^, ranging from 0.007 to 0.316, from 0.009 to 0.101, from 0.01 to 0.092, and from 0.092 to 2.44 mg kg^−1^, respectively. All 103 maize samples had Hg, 8 As, 91 Pb, and 36 Cr levels below the instrumental limit of detection (LOD_As_ = 0.001 μg L^−1^, LOD_Cr_ = 0.02 μg L^−1^). The mean values of ω(Cd), ω(As), ω(Pb), and ω(Cr) in the remaining maize samples were 0.032, 0.003, 0.056, and 0.07 mg kg^−1^, with ranges of 0.014–0.214, 0.001–0.009, 0.01–0.49, and 0.003–0 0.612 mg kg^−1^, respectively. A total of 15 cabbage samples had Hg, 3 As, 12 Pb, and 32 Cr below the limit of detection of the instrument. The mean values of ω(Cd), ω(Hg), ω(As), ω(Pb), and ω(Cr) in the remaining cabbage samples were 0.082, 0.002, 0.006, 0.141, and 0.039 mg kg^−1^, with ranges of 0.004–0.45, 0.001–0.003, 0.001–0.019, 0.011–1.38, and 0.034–0.046 mg kg^−1^, respectively.

The National Food Contaminant Limits Standard (GB 2762–2017) limits the maximum levels (MLs) of each heavy metal in food. Only two samples of cabbage in this study had Cd, and two samples had Pb concentrations exceeding the MLs (MLs for Cd = 0.2 mg kg^−1^; MLs for Pb = 0.3 mg kg^−1^). Four maize samples exceeded MLs for Cd and one for Pb (MLs for Cd = 0.1 mg kg^−1^; MLs for Pb = 0.2 mg kg^−1^1), and three rice samples had Cd and one sample had Cr levels exceeding MLs (MLs for Cd = 0.2 mg kg^−1^; MLs for Cr = 1.0 mg kg^−1^).

The order of variation coefficients was Pb > Cr > Cd > As in maize seeds, Cr > Cd > Pb > As in rice seeds, and Pb > Cd > As > Hg > Cr in cabbage. Cd and Pb in the three foods, as well as Cr in maize and rice, had very high coefficients of variation (CV > 1.0). This is consistent with the variability of the three metals in soil, which implies that the sources of heavy metals in foods in the study area are highly disturbed by external sources and may be related to anthropogenic activities such as mining. The skewness coefficients for Cd, Hg, As, Pb, and Cr in maize, rice, and cabbage were 1.036–4.224, 0.193–3.511, and 0.24–4.188, respectively. Except for Hg and As in rice, the skewness of each metal is greater than 1, which means that these elements have positive deviation values, and the data have “outlier data” on the right side of the mean, with more obvious deviation from the normal distribution. The kurtosis coefficients of Pb and Cr in rice; Cd, Pb, and Cr in maize; and Cd and Pb in cabbage were larger (>8), which indicates that the data for these elements were more concentrated compared with the normal distribution and had the largest difference from the normal distribution. The data for each heavy metal for the remaining crops were scattered.

### 3.3. Soil Contamination Evaluation

The geo−accumulation index (I_geo_) and potential ecological risk index (E_*r*_) were used to evaluate the soil heavy metal contamination in the study area (Figure 1). The results showed that the order of the three soil I_geo_ values in the study area was cabbage soil > maize soil > rice soil. The level of Cd contamination in the soils of the study area was high, and the levels of Hg, As, Pb, and Cr contamination were low. The order of magnitude of the mean I_geo_ for maize soils was Cd (1.35) > Hg (0.27) > Cr (0.1) > Pb (−0.03) > As (−0.47). In terms of the contamination levels of heavy metals, 5.83%, 33.01%, 35.92%, and 25.24% of the samples were at unpolluted (U), unpolluted to moderately polluted (U–M), moderately polluted (M), and moderately polluted to heavily polluted levels (M–H, H), respectively, for Cd. The contamination levels of Hg, As, Pb, and Cr were at unpolluted levels (U), and unpolluted to moderately polluted levels (U–M) were dominant. Compared with maize soils, the degree of contamination in rice soils was relatively low, and the order of the mean I_geo_ was Cd (−0.13) > Cr (−0.18) > Pb (−0.65) > Hg (−0.81) > As (−1.6). The contamination level of Cd was high, with half of the soil sites in the unpolluted to moderately and moderately polluted levels (U–M, M); the contamination levels of Hg, As, Pb, and Cr were relatively low. The percentage of soil samples with I_geo_ at the unpolluted level (U) was 86.36% for Hg, 90.91% for As, 81.82% for Pb, and 59.09% for Cr. The level of contamination in cabbage soil was higher than that in maize and rice soils, and the mean values of I_geo_ for heavy metals were greater than zero. The order of magnitude was Cd (5.51) > Pb (2.16) > Cr (0.77) > Hg (0.33) > As (0.09). More than 97% of the cabbage soils were evaluated for strong Cd contamination (H, H–E, E), with 62.86% of the soil samples reaching very strong contamination levels (E). In contrast, more than half of the soil samples for Hg, As, and Cr were below the moderately polluted level (M).

The results of the potential ecological risk (E_*r*_) evaluation are shown in Figure 1b. The mean E_*r*_ of Cd in the three soils was in the following order: cabbage soil (3903.51) >> maize soil (142.89) > rice soil (52.98). The potential ecological hazard of Cd in cabbage soils was the greatest, with 97.14% of Cd evaluated as being at an extremely high−risk level (E–H), and the E_*r*_ distribution of Cd in cabbage soils spanned a wide range of 318–19,410. The E_*r*_ range for Hg was 30.86–182.29, with 62.86% of Hg evaluated as medium risk (M) and 28.57% as high risk (H). Similar results were observed in maize soils with a Cd E_*r*_ range of 38.32–704.55, with 30.10%, 36.89%, and 25.24% for medium risk (M), high risk (H), and extremely high risk (E–H), respectively. The E_*r*_ range of Hg was 21.82–1043.64, and the percentages of medium risk (M) and high risk (H) were 48.54% and 31.07%, respectively. In rice soils, Cd and Hg were dominated by slight (S) and moderate risk (M) with 50% and 31.82% for Cd and 63.64% and 27.27% for Hg, respectively. The mean E_*r*_ values of As, Pb, and Cr in the three soils were less than 40, indicating a slight risk. In addition, except for Pb in cabbage soil, the mean values of E_*r*_ for all three soils were in the order of Cd > Hg > As > Pb > Cr. The mean values of the combined potential ecological risk index (R) for the three soils were in the order cabbage (4074.94) >> maize (262.77) > rice (104.44). A total of 62.85% of the cabbage soils had an R−value greater than 1200 and were evaluated as having an extremely high−risk level (E–H). Maize soils were predominantly medium risk (M) with 50.49%. Rice soils were less risky, with a slight risk level (S) of 17 samples, accounting for 77.27%.

### 3.4. Health Risk Assessment of Food Consumption

The US Environmental Protection Agency model (US EPA, 2011) was used to evaluate the carcinogenic risk (CR) and noncarcinogenic risk (THQ) of three foods consumed by adults and children in the local population (Figure 2). The mean values of CR for the consumption of the three crops are higher for Cr and As, which may pose a potential carcinogenic risk to humans after excessive intake of Cr and As [45]. Long−term intake of Cr may cause flat epithelial cancer, adenocarcinoma, and lung cancer [46]. The mean CR for Cr after consumption of the three crops was 6.44 × 10^−4^ and 2.78 × 10^−4^ for children and adults, respectively, which exceeds the unacceptable threshold (1 × 10^−4^), implying that 644 per 1 million children and 278 per 1 million adults are at risk of cancer due to Cr intake. Chronic As poisoning can be caused by the intake of food containing As, which is carcinogenic, teratogenic, and mutagenic to humans [47]. The mean CR of As in rice was greater than 1 × 10^−4,^ whereas the mean CR of As in maize and cabbage was less than 1 × 10^−4^. This may be due to the high bioaccumulation of As by rice. The mean CR values of Cd, Pb, and Hg in all three crops were less than 1 × 10^−4^, indicating that their risks would not cause significant effects. The order of the mean CR of heavy metals in the three crops was Cr > As > Pb > Cd > Hg. The order of total cancer risk (TCR) for consumption of the three crops was rice > maize > cabbage. Children were exposed to a higher cancer risk than adults. The major contributor to TCR was Cr, with a contribution of more than 66%.

The noncarcinogenic risk results are shown in Figure 3. Children consume less food than adults; however, children have higher heavy metal hazard quotient (THQ) values than adults. The THQ values were 1.91 and 3.22 for adults and children in rice, 0.37 and 0.96 for adults and children in cabbage, and 0.2 and 0.51 for adults and children in maize, respectively. According to the US EPA recommendations, the average HQ for children and adults due to all heavy metals should be less than 1.0 [48]. Our study showed that the THQ values for both adults and children were greater than 1 in rice, which poses a health risk to humans, and less than 1.0 in both adults and children in cabbage and corn. High noncarcinogenic risk (THQ > 1) was found in 28.75% and 13.13% of children and adults in the study area, respectively. Similar to heavy metal contamination in soil, Cd has the highest HQ among the five heavy metals, which means that people are at health risk due to Cd intake. The order of the THQ values for the three foods in the study area was rice > cabbage > maize. For rice, As and Cr were the major contributors to THQ, accounting for 47.13% and 32.15%, respectively. For cabbage, Cd was the main contributor to THQ, with a contribution of 53.31%. The order of the contribution of THQ in maize was Cd (38.63%) > Cr (28.52%) > As (19.48%) > Pb (13.36%).

### 3.5. Prediction Model of Metal Transfer from Soil to Crops

To predict heavy metal concentrations in foods, we established a regression equation based on the logarithm of heavy metal concentrations in soil and soil pH and the logarithm of heavy metal concentrations in foods (Table 5). It can be seen that the Cd concentrations in the three foods can be well−predicted, with all equations reaching a significant level with *p*−values less than 0.005. However, the predictions for the other heavy metals (Hg, As, Pb, and Cr) were unsatisfactory (not listed), and none of them reached significant levels. This may be related to the low concentrations of Hg, As, Pb, and Cr in the soils of the study area. In particular, the concentrations of Hg, As, Pb, and Cr in more samples from plants were below the instrumental limits of detection, and these reasons increased the model error. For Cd, the predictions of Cd concentrations in the three foods were in good agreement with the mean or range observations, especially for leafy cabbage (Table 5 and Figure 4), implying that the model simulations of Cd concentrations in rice, maize, and cabbage were reasonable. However, the model predictions for maize and rice slightly underestimated the concentrations. In addition, four maize samples here had Cd concentrations above the MLs (0.1 mg kg^−1^), but predictions indicated that the Cd concentrations in maize in the study area were below the threshold. This implies that maize was severely disturbed by Cd contamination throughout the study area [49].

## 4. Conclusions

The soils of maize, rice, and cabbage in the study area showed high accumulation characteristics of heavy metals, with average concentrations exceeding the national background values by 1.6–130 times, with Cd contamination being the most serious. The results of the geo−accumulation index evaluation showed that the percentage of samples with Cd at the moderate polluted level or below in maize soils was as high as 74.76%; half of the rice soil sites were at the unpolluted level; and more than 97% of the cabbage soils had Cd pollution above the heavily polluted level. The order of the combined potential ecological risk index was cabbage (4074.94) >> maize (262.77) > rice (104.44). Cd was the major contributor, with 95.79% for cabbage, 54.38% for maize, and 50.73% for rice. The results of the risk assessment of agricultural product consumption indicated a high carcinogenic and noncarcinogenic risk for the local population. The top contributor to cancer risk was Cr, followed by As. Cd was the main contributor to noncarcinogenic risk in maize and cabbage, and in rice, it was mainly caused by As. The risk was higher in children than in adults and was the highest for rice consumption. In addition, the predictive models for heavy metals showed sufficient predictability of Cd levels in foods and reasonable simulations of Cd concentrations in rice, maize, and cabbage throughout the study area. It could allow decision−making on the need for remediation strategies to reduce the risk of metal contamination of agricultural land in potentially high−risk areas of karst.

## Figures and Tables

**Figure 1 foods-11-02802-f001:**
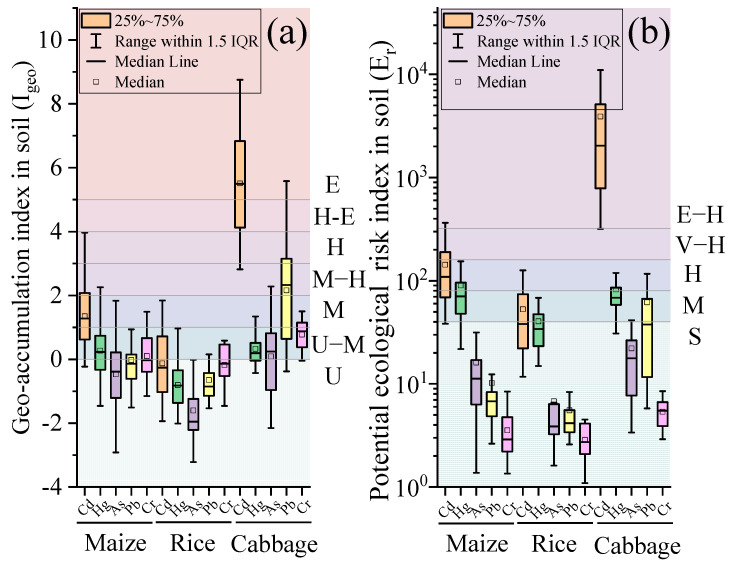
Values of geo−accumulation index (I_geo_) (**a**) and single pollution index in crops (*P_j_*) (**b**) of Cd, Hg, As, Pb, and Cr. Note: in (**a**), U represents unpolluted, U−M represents unpolluted to moderately polluted, M represents moderately polluted, M−H represents moderately to heavily polluted, H represents heavily polluted, H−E represents heavily polluted to extremely polluted, and E represents extremely polluted; in (**b**), S represents slight risk, M represents medium risk, H represents high risk, V−H represents very high risk, and E−H represents extremely high risk.

**Figure 2 foods-11-02802-f002:**
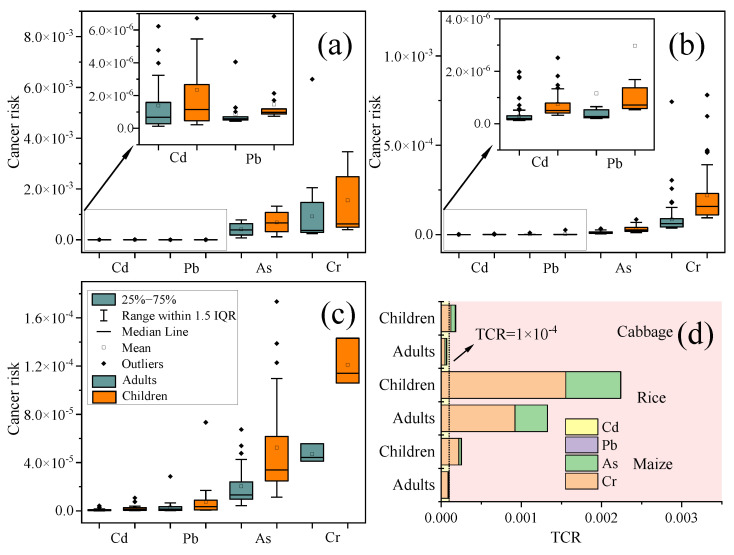
Carcinogenic health risks (CR) in study area from consumption of (**a**) rice, (**b**) maize, and (**c**) cabbage. Total carcinogenic health risks (TCR) (**d**) in study area. Orange boxes represent CR for children, and cyan boxes represent CR for adults.

**Figure 3 foods-11-02802-f003:**
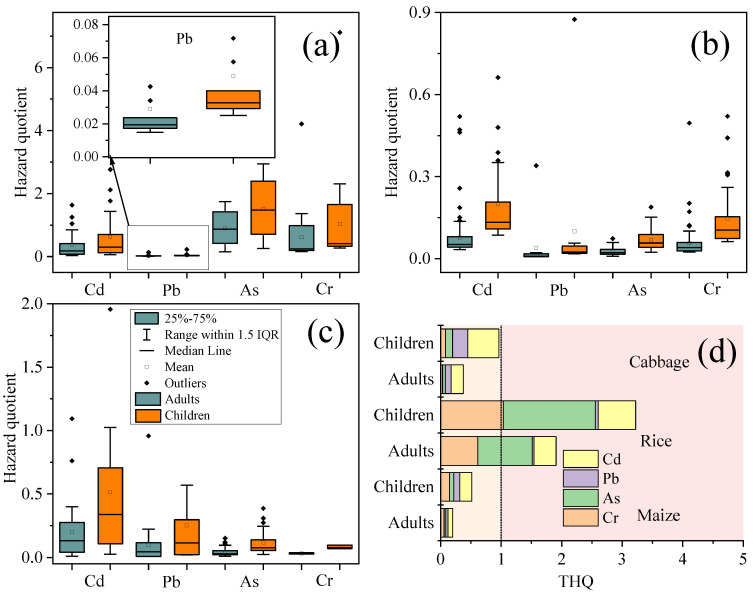
Hazard quotient (HQ) in study area from consumption of (**a**) rice, (**b**) maize, and (**c**) cabbage. Total carcinogenic health risks (THQ) (**d**) in study area. Orange boxes represent HQ for children, and cyan boxes represent HQ for adults.

**Figure 4 foods-11-02802-f004:**
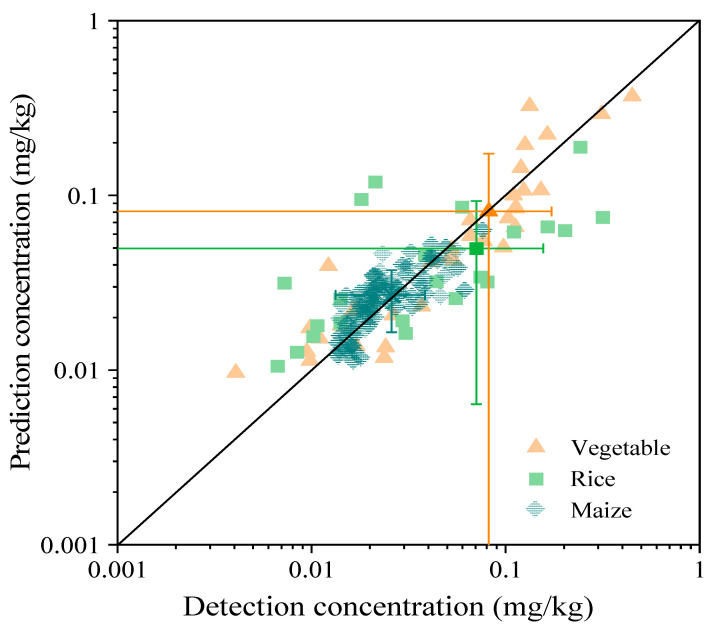
Model simulation results for Cd (note: 0.0001 has been added to all values to avoid values of 0, the line is a 1:1 line, and error bars indicate the maximum and minimum values).

**Table 1 foods-11-02802-t001:** Contamination classification of I_geo_.

Class	Range	Classification
0	I_geo_ < 0	Uncontaminated
1	0 ≤ I_geo_ < 1	Uncontaminated to moderately contaminated
2	1 ≤ I_geo_ < 2	Moderately contaminated
3	2 ≤ I_geo_ < 3	Moderately to heavily contaminated
4	3 ≤ I_geo_ < 4	Heavily contaminated
5	4 ≤ I_geo_ < 5	Heavily to extremely contaminated
6	I_geo_ > 5	Extremely contaminated

**Table 2 foods-11-02802-t002:** Grades of potential ecological risk index.

Potential Risk Level	E_*r*_	R
Slight risk	E_*r*_ < 40	R < 150
Moderate risk	40 ≤ E_*r*_ < 80	150 ≤ R < 300
High risk	80 ≤ E_*r*_ < 160	300 ≤ R < 600
Very high risk	160 ≤ E_*r*_ < 320	600 ≤ R < 1200
Extremely high Risk	E_*r*_ ≥ 320	R ≥ 1200

**Table 3 foods-11-02802-t003:** Characteristic statistics of pH values and heavy metal concentrations in soil from agricultural soils in the research area.

		Soil pH	Heavy Metals in Soil (mg kg^−1^)				
			Cd	Hg	As	Pb	Cr
Rice soil (*n* = 22)	Min	4.67	0.26	0.04	3.23	18.20	52.30
	Max	7.47	3.55	0.32	51.90	102.00	216.00
	Mean	6.05	1.17	0.11	13.57	39.01	137.57
	SD	0.74	0.89	0.08	13.83	25.49	52.63
	CV	0.12	0.76	0.68	1.02	0.65	0.38
	1st quartile	5.47	0.49	0.06	6.48	23.90	100.00
	Median	6.05	0.84	0.09	7.74	29.30	131.00
	3rd quartile	6.78	1.63	0.13	12.70	39.20	198.00
	Skewness	0.08	1.34	1.96	2.16	1.79	0.14
	Kurtosis	−0.91	1.16	3.61	3.86	2.18	−1.01
Maize soil (*n* = 103)	Min	4.53	0.30	0.06	2.74	18.50	61.30
	Max	8.09	15.50	2.87	532.00	696.00	404.00
	Mean	6.35	2.96	0.25	31.75	71.69	168.98
	SD	0.93	2.47	0.30	53.85	97.88	83.16
	CV	0.15	0.83	1.20	1.70	1.37	0.49
	1st quartile	5.64	1.38	0.13	12.40	33.10	99.60
	Median	6.30	2.21	0.19	22.50	47.80	137.00
	3rd quartile	7.06	3.61	0.26	34.10	58.60	228.00
	Skewness	0.18	2.50	7.14	8.08	4.55	0.93
	Kurtosis	−0.87	8.29	61.50	74.49	23.64	0.15
Cabbage soil (*n* = 35)	Min	4.35	1.06	0.05	3.78	30.00	88.70
	Max	7.77	64.70	0.32	81.70	1864.00	259.00
	Mean	6.13	13.01	0.14	24.77	320.60	163.83
	SD	0.93	16.31	0.07	20.76	429.59	50.06
	CV	0.15	1.25	0.47	0.84	1.34	0.31
	1st quartile	5.25	2.62	0.10	8.63	60.70	119.00
	Median	6.12	6.78	0.12	19.90	196.00	168.00
	3rd quartile	7.11	17.10	0.15	29.60	347.00	203.00
	Skewness	0.12	2.02	1.40	1.59	2.63	0.25
	Kurtosis	−0.95	3.47	1.05	2.31	6.84	−1.13
National background value in soil		/	0.10	0.07	11.20	26.00	61.00

**Table 4 foods-11-02802-t004:** Descriptive statistics of heavy metal concentrations in food samples in the study area.

		Heavy Metals in Soil (mg kg^−1^)				
		Cd	Hg	As	Pb	Cr
Rice (*n* = 22)	Numbers of ND	0	22	0	7	0
	Min	0.007	/	0.009	0.010	0.092
	Max	0.316	/	0.101	0.092	2.440
	Mean	0.071	/	0.052	0.020	0.355
	SD	0.085	/	0.031	0.021	0.514
	CV	1.209	/	0.589	1.050	1.449
	1st quartile	0.014	/	0.024	0.012	0.113
	Median	0.034	/	0.051	0.013	0.143
	3rd quartile	0.080	/	0.082	0.016	0.568
	Skewness	1.752	/	0.193	3.511	3.472
	Kurtosis	2.423	/	−1.274	12.856	13.706
Maize (*n* = 103)	Numbers of ND	0	103	8	91	36
	Min	0.014	/	0.001	0.010	0.030
	Max	0.214	/	0.009	0.490	0.612
	Mean	0.032	/	0.003	0.056	0.070
	SD	0.033	/	0.002	0.137	0.079
	CV	1.029	/	0.486	2.442	1.123
	1st quartile	0.017	/	0.002	0.011	0.035
	Median	0.021	/	0.003	0.014	0.050
	3rd quartile	0.033	/	0.004	0.026	0.074
	Skewness	4.224	/	1.036	3.446	5.329
	Kurtosis	19.421	/	0.793	11.908	34.319
Cabbage (*n* = 35)	Numbers of ND	0	15	3	12	32
	Min	0.004	0.001	0.001	0.011	0.034
	Max	0.450	0.003	0.019	1.380	0.046
	Mean	0.082	0.002	0.006	0.141	0.039
	SD	0.090	0.001	0.004	0.282	0.006
	CV	1.103	0.340	0.791	2.001	0.162
	1st quartile	0.017	0.001	0.003	0.013	0.034
	Median	0.054	0.002	0.004	0.064	0.037
	3rd quartile	0.113	0.002	0.007	0.166	0.046
	Skewness	2.500	0.240	1.406	4.188	1.402
	Kurtosis	8.029	−0.906	1.224	18.811	/

**Table 5 foods-11-02802-t005:** Relationship between metal content in foods (mg kg^−^^1^) and soil pH and soil total metal concentrations (mg kg^−^^1^).

Plant Type	Heavy Metals	Regression Equation	*n*	R^2^	*p*	Detection Value	Prediction Value
Cabbage	Cd	C_f_ = 10^(−1.845 − 0.039 × pH + 0.912 × $\log_{10}^{\left[\mathrm{Cs}\right]}$)	35	0.850	<0.001	0.082 (0.004–0.450)	0.081 (0.009–0.370)
Maize	Cd	C_f_ = 10^(−0.748 − 0.151 × pH + 0.322 × $\log_{10}^{\left[\mathrm{Cs}\right]}$)	102	0.504	<0.001	0.032 (0.014–0.210)	0.028 (0.011–0.073)
Rice	Cd	C_f_ = 10^(−0.133 − 0.208 × pH + 1.025 × $\log_{10}^{\left[\mathrm{Cs}\right]}$)	22	0.438	<0.005	0.071 (0.006–0.316)	0.050 (0.011–0.189)

## Data Availability

Data sharing not applicable to this article as no datasets were generated or analysed during the current study.

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
