# Peer review of "Characteristics, Health Risk Assessment, and Transfer Model of Heavy Metals in the Soil—Food Chain in Cultivated Land in Karst"

_foods, 2022, doi:10.3390/foods11182802_

Round 1

Reviewer 1 Report

The paper aimed at determining the concentrations of some heavy metals (Cd, Hg, As, Pb, and Cr) in agricultural soils and crops (rice, maize, and cabbage) in Shuicheng District of Liupanshui City (Guizhou Province) as well as at evaluating the potential health risks to adults and children from consuming the three crops. The order of potential risk of heavy metals was Cd > Hg > As > Pb > Cr.

The study is interesting, the subject is topical in the related field, the methodology is adequate and explicitly stated, there is no redundant information, the results are significant, the conclusions are adequately supported by the data presented, but the text should be carefully revised and some sentences reformulated. I suggested some modifications in the attached document.

Author Response

Replies to Reviewer 1:

The paper aimed at determining the concentrations of some heavy metals (Cd, Hg, As, Pb, and Cr) in agricultural soils and crops (rice, maize, and cabbage) in Shuicheng District of Liupanshui City (Guizhou Province) as well as at evaluating the potential health risks to adults and children from consuming the three crops. The order of potential risk of heavy metals was Cd > Hg > As > Pb > Cr.

The study is interesting, the subject is topical in the related field, the methodology is adequate and explicitly stated, there is no redundant information, the results are significant, the conclusions are adequately supported by the data presented, but the text should be carefully revised and some sentences reformulated. I suggested some modifications in the attached document.

Answer: Thanks for your valuable advice and recommendation. We have carefully revised those sentences followed your comments.

Reviewer 2 Report

Introduction

The authors give a descriptive introduction which also highlights the problem concerning the lack of information regarding heavy metal transfer to foods in these areas. The authors also decided to focus on (Cd, Hg, As, Pb and Cr) which are often seen as the heavy metal with the most detrimental impact on environments, plant growth, development and plant nutrition.

Methods

The methods were well defined and easy to follow.

Results

From the data presented it can be observed that an increase in pH leads to an increase in heavy metal deposition/concentrations for all soils (cabbage, rice and maize). I can’t remember this being mentioned in the results text. This may be an important factor to mention as it would help the ultimate aim of the article to advise policy makers regarding potential risk and remediation strategies.

Figure were clear and convey their message.

Line 269-270: Were some of the soils not accounted for when the classifying for contamination levels? The percentages given do not equal to 100%

Line 272: “Cd. Hg, As,” add a comma after Cd

Line 320. “which is above the unacceptable threshold” Is it suppose to be unacceptable threshold or acceptable?

Line 3118-321: The authors start the sentence concerning Cancer risk for Cr but then associates cancer risk to Cd (line 321). Please look at this sentence again.

The authors give a comprehensive conclusion concerning the most important findings and how they may impact the local population.

Author Response

Replies to Reviewer 2:

  • Line 269-270: Were some of the soils not accounted for when the classifying for contamination levels? The percentages given do not equal to 100%

Answer: Thanks for your valuable advice and recommendation. We have changed the data in line 270

  • Line 272: “Cd. Hg, As,” add a comma after Cd

Answer: Thanks for your valuable advice and recommendation. This is due to the incompleteness of the latter sentence. We have made adjustments in line 273.

  • Line 320. “which is above the unacceptable threshold” Is it suppose to be unacceptable threshold or acceptable?

Answer: Thanks for your valuable advice and recommendation. It is unacceptable threshold (1 × 10‒4), and we have made adjustments in line 316.

  • Line 318-321: The authors start the sentence concerning Cancer risk for Cr but then associates cancer risk to Cd (line 321). Please look at this sentence again.

Answer: Thanks for your valuable advice and recommendation. We have changed the word in line 323